# Assessment of the Genetic Diversity and Population Structure of *Rhizophora apiculata* Blume (Rhizophoraceae) in Thailand

**DOI:** 10.3390/biology11101449

**Published:** 2022-10-01

**Authors:** Panthita Ruang-areerate, Chaiwat Naktang, Wasitthee Kongkachana, Duangjai Sangsrakru, Nattapol Narong, Chatree Maknual, Tamanai Pravinvongvuthi, Waratthaya Promchoo, Suchart Yamprasai, Sithichoke Tangphatsornruang, Wirulda Pootakham

**Affiliations:** 1National Omics Center, National Science and Technology Development Agency (NSTDA), Pathum Thani 12120, Thailand; 2Department of Marine and Coastal Resources, 120 The Government Complex, Chaengwatthana Rd., Thung Song Hong, Bangkok 10210, Thailand

**Keywords:** mangrove, *Rhizophora apiculata*, Rhizophoraceae, whole-genome, genetic diversity, population structure, SNP

## Abstract

**Simple Summary:**

We utilized the 10× Genomics technology to obtain a reference whole-genome sequence for assessing the genetic diversity and population structure of *Rhizophora apiculata* in Thailand. Using SNPs identified from the *R. apiculata* genome sequence, moderate genetic diversity and high genetic differentiation were observed among 82 *R. apiculata* accessions collected along the coasts of Thailand. Two subpopulations corresponding to the Gulf of Thailand and the Andaman Sea coasts were clustered and confirmed from three approaches: population structure, PCA, and phylogenetic analyses. The AMOVA result revealed that the percentage of variation within populations (76%) was higher than that among populations (24%).

**Abstract:**

*Rhizophora apiculata* is one of the most widespread and economically important mangrove trees in the Indo-West Pacific region. Knowledge of the genetic variation of *R. apiculata* in Thailand is limited. Here, we generated a whole-genome sequence of *R. apiculata* using the 10× Genomics technology. *R. apiculata* genome assembly was 230.47 Mb. Based on its genome, 2640 loci of high-quality biallelic SNPs were identified from 82 *R. apiculata* accessions collected from 17 natural mangrove forests in Thailand to assess the genetic diversity and population structure among them. A moderate level of genetic diversity of *R. apiculata* was observed. The average observed heterozygosity (*H*o = 0.48) was higher than the average expected heterozygosity (*H*e = 0.36). Two subpopulations were observed and confirmed from three approaches: population structure, PCA, and phylogenetic analyses. They corresponded to the Gulf of Thailand and the Andaman Sea separated by the Malay Peninsula. AMOVA analyses indicated that genetic variation was attributable to 76.22% within populations and 23.78% among populations. A high level of genetic differentiation between the two subpopulations (*F*_ST_ = 0.24, *p* < 0.001) was observed. This study evaluated the genetic diversity and population structure of *R. apiculata*, providing useful information for sustainable mangrove management in Thailand.

## 1. Introduction

Mangroves generally grow in intertidal habitats, which are at the interface between land and sea in tropical and sub-tropical regions [1]. They are the most important component of coastal ecosystems and ecological services [2,3,4,5,6]. Mangrove trees protect against coastal erosion, reduce the effects of strong winds and heavy waves on the coasts as well as provide habitats, food, timber, and medicines [5,6]. In addition, mangrove trees can serve as carbon sinks that mitigate greenhouse effects [2,3,4]. In the past decades, coastal regions have been changed by global climate change and anthropogenic activities (e.g., aquaculture encroachment and urban area extension) that affect the loss of habitat and genetic variation of mangroves [7,8,9]. In Thailand, mangrove forest areas along the coasts of the Gulf of Thailand and the Andaman Sea have dramatically decreased from 367,900 to 245,179 hectares in 1961 and 2000, respectively [10,11]. As a result, it is important to implement strategies to protect, preserve, and reforest mangrove areas [12]. The study of the genetic diversity and population structure of mangroves is important to manage rehabilitation and sustainability in the future.

*Rhizophora apiculata* belongs to the family Rhizophoraceae and is a true mangrove that is one of the most widespread and economically important plant species in tropical regions [13]. It is distributed in the Indo-West Pacific (IWP) region, in countries such as India, Indonesia, Malaysia, Myanmar, and Thailand [13,14]. It is used to make firewood and charcoal [15]. In addition, its leaves, barks, and roots have medicinal uses, such as antimicrobial, anticancer, antidiarrhea, and hemostatic properties [16,17,18]. Based on the morphological structure of *R. apiculata*, the leaf shape is narrowly elliptic-oblong and the bark is dark grey and vertically fissured [15]. Remarkably, prop and stilt roots (aerial roots) are common in *Rhizophora* species (e.g., *R. apiculata*, *R. mucronata*, *R. mangle*, and *R. stylosa*) for supporting respiration when their lower roots are submerged and elevating the plants above the water. [19,20]. Notably, *R. apiculata* is one of the most regenerated mangrove species in Thailand because seeds are easy to plant in nurseries [12,21].

Molecular markers, such as amplified fragment length polymorphisms (AFLPs), nuclear and chloroplast DNA sequences, microsatellites, and single nucleotide polymorphisms (SNPs), have been used to study population genetics in mangroves [22,23,24,25,26,27,28,29,30,31,32]. For evaluating genetic diversity and population genetic structure, microsatellites are widely used [24,26,28,29]; however, SNPs generated from next-generation sequencing approaches, such as genotyping by sequencing (GBS) and restriction site-associated DNA sequencing (RAD-seq) [33], have recently become a favorable molecular marker in plants due to high-throughput genetic data and the most abundant genetic polymorphisms across genomes [27,30,31,32]. Several studies on genetic diversity and structure have been conducted on *R. apiculata* populations using various molecular markers [23,24,25,26,27,28,29]. For example, an earlier assessment of 31 *R. apiculata* individuals in three collection sites (Bangkok, Surat Thani, and Trang) in Thailand revealed low genetic variation and a significant genetic differentiation between populations based on five nuclear genes and two chloroplast DNA regions [23]. Population structure consists of two clusters, Bangkok-Surat Thani and Trang [23]. In contrast to the Indo-Malaysian region, eleven *R. apiculata* populations based on 81 nuclear loci showed high genetic diversity and various levels of genetic differentiation among populations [25]. They were separated into three clusters; East Indian Ocean, South China Sea, and Australasia [25]. Using microsatellite markers, the genetic diversity and population structure of *R. apiculata* populations were studied in several regions, such as the Greater Sunda Islands of Indonesia [24], the IWP region [26], Malaysia [28], and China [29]. The genetic diversity of *R. apiculata* was low in the IWP region and Malaysia but high in China [26,28,29]. High genetic differentiation between populations was found in the Greater Sunda Islands of Indonesia, the IWP region, and Malaysia [24,26,28]. Intermediate genetic diversity was revealed between the Hainan Island-Gulf of Thailand and the west coast of Thailand based on SNP data [27].

The genetic diversity and population structure of *R. apiculata* across the range of Thailand coasts has not been evaluated. To obtain a whole-genome reference sequence *R. apiculata* in Thailand, the whole genome of *R. apiculata* was sequenced and assembled as a reference sequence for this study. Based on our *R. apiculata* reference genome, numerous variants were identified from 82 *R. apiculata* accessions collected from 17 natural mangrove forests in Thailand. The variants were filtered to identify high-quality biallelic SNPs that were used to assess the genetic diversity and population structure of *R. apiculata* in Thailand.

## 2. Materials and Methods

### 2.1. Samples

For reference genome sequencing, one *R. apiculata* individual was chosen as a representative species in this study. It is in the natural mangrove forest in the Ranong province (9°52′36.1″ N 98°36′11.5″ E) under the protection of the Department of Marine and Coastal Resources. The morphology of *R. apiculata* as a reference genome is presented in Figure 1.

For genetic diversity and population structure analysis, a total of 82 *R. apiculata* accessions were collected from natural mangrove forests in 17 provinces of Thailand (Figure 2). The seventeen collection sites were Chachoengsao (CCO), Chumphon (CMP), Chanthaburi (CTI), Nakhon Si Thammarat (NST), Narathiwat (NWT), Phetchaburi (PBI), Prachuap Khiri Khan (PKN), Phuket (PKT), Phang-nga (PNA), Pattani (PTN), Ranong (RNG), Samut Sakhon (SKN), Surat Thani (SNI), Samut Prakan (SPK), Satun (STN), Trang (TRG), and Trat (TRT). The collection sites of CCO, CMP, CTI, NST, NWT, PBI, PKN, PTN, SKN, SNI, SPK, and TRT represent locations on the Gulf of Thailand coast. In addition, the collection sites of PKT, PNA, RNG, STN, and TRG represent locations on the Andaman Sea coast. These two coastal regions were regionally separated by the Malay Peninsula. The geographic map was created to locate collection sites using the QGIS software v3.24.2 [34]. The general characters of *R. apiculata* accessions have large and tall trees, a narrowly elliptic leaf shape, brown to dark grey bark, and prop and stilt roots.

### 2.2. DNA Extraction and Sequencing

The fresh leaves of all *R. apiculata* accessions were stored in liquid nitrogen. The standard CTAB (Cetyl Trimethyl Ammonium Bromide) method was used to extract genomic DNA from the *R. apiculata* leaves [35]. One *R. apiculata* accession as a reference sequence was sequenced using the 10× Genomics technology with linked-read sequencing, which is a microfluidics-based method to generate long-range information from short-read sequencing data (10× Genomics; https://www.10xgenomics.com (accessed on 5 January 2022)). The 10× genomics library was constructed from approximately 1 ng of high quality, high molecular weight DNA using the Chromium Genome Library Kit & Gel Bead Kit v2, the Chromium Genome Chip Kit v2 and the Chromium i7 Multiplex Kit according to the manufacturer’s instructions (10× Genomics) and was sequenced using the Illumina HiSeq X Ten (paired-end reads at 150 bp). Furthermore, the 82 *R. apiculata* accessions were sequenced using MGI technology with restriction site-associated DNA sequencing (RAD-seq), which is one of the most genomic library preparations to reduce representation sequencing [33]. For each sample, a RAD-seq library was created with approximately 1 µg of DNA following the MGIEasy RAD Library Prep Kit Instruction Manual (MGI Tech). Libraries were pooled and sequenced using the MGISEQ-2000RS sequencing platform with a 150 bp paired-end cycle kit following the manufacturer’s protocol. High-quality reads were obtained from the MGISEQ-2000RS sequencer.

### 2.3. Genome Assembly and Comparative Genome Analysis

The whole genome of *R. apiculata* was assembled to link read data using SuperNova v2.1.1 with default settings (https://support.10xgenomics.com/de-novo-assembly/software/pipelines/latest/using/running (accessed on 10 January 2022); 10× Genomics; [36]). All contigs were scaffolded with the previously reported *R. apiculata* genome (GCA_900174605.1 in the European Nucleotide Archive (ENA), [37]) using RagTag v1.1.0 [38]. Redundant and mitochondrial contigs were removed using BLAST with 100% identity and 100% coverage [39]. Our *R. apiculata* genome was assessed with the previous *R. apiculata* genome [37] using QUAST and Busco [40,41]. The genome assembly of *R. apiculata* has been deposited in the National Center for Biotechnology Information (NCBI) under the project accession number PRJNA846534.

### 2.4. SNP Identification

To identify single nucleotide polymorphisms (SNPs), paired-end reads of each accession were mapped to our genome sequences of *R. apiculata* using BWA v0.7.17-r1188 (https://github.com/lh3/bwa (accessed on 2 February 2022)) with default settings. Sequence alignment map (SAM) files were converted to a binary format (BAM) that were sorted and indexed using samtools v1.9 [42]. Mapping statistics were also obtained using samtools. Variants were called from each sorted BAM file using GATK v4.1.4.1 with haplotype caller [43]. Individual variant call format (VCF) files were merged to a single VCF file using GATK v4.1.4.1. All variants, including SNPs and indels (insertions and deletions) of accessions, were called from the single VCF file, and only SNPs were extracted using GATK. Multiallelic sites of SNPs were excluded from further analysis using bcftools v1.12 [44]. SNPs were filtered using vcftools v0.1.16 [45] with a minimum read depth ≥ 10, maximum read depth < 200, missing data < 0.05, and minor allele frequency (MAF) ≥ 0.05 [46]. Finally, SNPs on scaffolds were selected for further analysis.

### 2.5. Population Structure and Principal Component Analysis

The population structure analysis of *R. apiculata* was examined using the STRUCTURE program v2.3.4 with a Bayesian clustering approach via the Markov Chain Monte Carlo (MCMC) estimation [47,48]. To prepare input data for STRUCTURE, the final VCF file was converted to the STRUCTURE file format using PGDSpider v2.0.9.2 [49]. Then, the analysis was performed using twenty replicate runs in each number of clusters (K) from 1 to 10, an MCMC burn-in period length of 10,000, and a run length of 10,000 [32]. The most probable K value (the number of subpopulations) was determined by comparing delta K (∆K) based on the rate of change in the log-probability of the data [LnP(D)] between successive K values using the Structure Harvester v0.6.7 [50,51]. Based on the best K value, clustering from 1000 replicates in STRUCTURE was summarized using CLUMPP version 1.1.2 [52].

To determine the proportion of variation explained by each principal component, eigenvalues of the filtered SNPs were generated using PLINK v1.9 [53]. Principle component analysis was performed using PCA in R [54]. The first and second principal components were plotted using R software v3.3.4 with the library tidyverse and the package ggplot [55].

### 2.6. Data Analysis

Gene diversity (heterozygosity), polymorphism information content, and minor allele frequency were carried out using PowerMarker v3.25 [56]. Additionally, basic genetic diversity parameters, including the effective number of alleles (*N*e), Shannon’s information index (*I*), observed (*H*o) and expected (*H*e) heterozygosity, the percentage of polymorphic loci (PPL), and the inbreeding coefficient (*F*), were generated using GenAlEx v6.502 [57]. The analysis of molecular variance (AMOVA) and the genetic differentiation coefficient (*F*_ST_) was conducted using Arlequin v3.5.2.2 [58]. An *F*_ST_ value was greater than 0.15, indicating the genetic differentiation between populations [59].

### 2.7. Phylogenetic Analysis

To assess the phylogenetic relationships among the 82 *R. apiculata* accessions, phylogenetic analysis was performed using a maximum likelihood (ML) method based on SNPs. The SNPs in all accessions were aligned using MUSCLE with default in MEGA X [60]. To be the best fit model for the SNP dataset, the K2 + G model was identified using the find best DNA/protein model tool in MEGA X. A ML phylogenetic tree was constructed using MEGA X with the K2 + G model. A bootstrap consensus tree with 1000 replications was carried out. The phylogenetic tree was visualized using the interactive Tree of Life (iTOL) [61].

## 3. Results

### 3.1. Genome Assembly and SNP Data

For the whole-genome sequencing of *R. apiculata*, a total of 110.04 Gb of 150 bp paired-end reads were obtained and assembled. The de novo assembly contained 22,267 contigs with the longest scaffold, N50 length, and N90 length of 1,917,847, 144,278, and 3785 bp, respectively (Appendix A). The contigs were further scaffolded based on the previous *R. apiculata* genome by merging homology sequences and reconciling genome assembly scaffolds [37]. The final *R. apiculata* genome assembly contained 133 scaffolds and 10,427 contigs with an aggregate size of 230.47 Mb (Appendix A). The longest scaffold, N50, and N90 were 12,839,107; 4,834,853; and 74,632 bp in length, respectively (Appendix A). The size of our *R. apiculata* genome (231 Mb) is similar to the size of the previously reported *R. apiculata* genomes (232 Mb) in China [37,62], which covered 84% of the estimated *R. apiculata* genome based on flow cytometry (274 Mb) [37]. The BUSCO (benchmarking universal single-copy orthologues) result revealed that all scaffolds (208 Mb, Appendix A) covered approximately 97% of predicted *R. apiculata* genes (Appendix A) [37].

To generate SNPs data, 82 *R. apiculata* accessions in natural mangrove forests in 17 provinces of Thailand (Figure 2) were sequenced using RAD-seq. A total of 226.04 Gb of 150 bp paired-end reads were generated (Appendix A). A total of 1,745,767 SNPs were identified among 82 *R. apiculata* accessions. Numerous SNPs with missing data > 0.05 (1,395,365 SNPs) and minimum read depth < 10 in each accession (313,981 SNPs) were removed. After filtering SNPs using the criteria mentioned in the Materials and Methods Section, 2640 high-quality biallelic SNPs were obtained.

### 3.2. SNP Characterization

The distributions of values for genetic diversity, polymorphic information content (PIC), and minor allele frequency (MAF) of the 2640 SNPs estimated on the 82 *R. apiculata* accessions are shown in Figure 3. The SNP diversity ranged from 0.16 to 0.50 with an average of 0.39 with the vast majority (96%) falling between 0.21 and 0.50 (Figure 3A; Appendix A). The PIC values ranged from 0.15 to 0.38 with a mean PIC value of 0.31 (Figure 3B; Appendix A). Approximately 60% of the SNPs had PIC values exceeding 0.30, suggesting high polymorphic data. The MAF values varied from 0.09 to 0.50, with an average value of 0.31 (Figure 3C; Appendix A). 

### 3.3. Population Structure and Principal Component Analysis

To classify subpopulations, the population structure and principal component analysis (PCA) of *R. apiculata* in Thailand was performed (Figure 4). The Bayesian clustering algorithm was used to analyze the population structure. The largest delta K was observed at K = 2 (Figure 4A), suggesting the presence of two subpopulations. The first subpopulation (orange in Figure 4B) consists of accessions collected from TRT, CTI, CCO, SPK, SKN, PBI, PKN, CMP, SNI, NST, PTN, and NWT, and the second subpopulation (blue in Figure 4B) consists of accessions from STN, TRG, PKT, PNA, and RNG. The analysis demonstrated two distinct genetic clusters corresponding to two geographic regions, the Gulf of Thailand (TRT, CTI, CCO, SPK, SKN, PBI, PKN, CMP, SNI, NST, PTN, and NWT) and the Andaman Sea (STN, TRG, PKT, PNA, and RNG). In addition to the population structure, the PCA of *R. apiculata* was conducted across the 82 *R. apiculata* accessions (Figure 4C) where the first two components explained 70.46% of the total genetic variation (67.60% and 2.86%, respectively). According to the first two components, the accessions were divided into two groups: the Gulf of Thailand (TRT, CTI, CCO, SPK, SKN, PBI, PKN, CMP, SNI, NST, PTN, and NWT) and the Andaman Sea (STN, TRG, PKT, PNA, and RNG).

### 3.4. Genetic Diversity and Genetic Differentiation

At the population level, the average number of effective alleles (*N*e), Shannon’s information index (*I*), observed heterozygosity (*H*o), expected heterozygosity (*H*e), and the percentage of polymorphic loci (PPL) were estimated (Table 1; Appendix A). All diversity parameters were similar in the two subpopulations. The PPL value in the two subpopulations was over 99%, indicating high genetic diversity within them. Remarkably, average *H*o values were higher than average *H*e values, which resulted in a negative inbreeding coefficient value (*F* = −0.199), indicating an excess of heterozygosity.

To assess genetic differences, AMOVA (analysis of molecular variance) analyses of the accessions with the two subpopulations revealed that 23.78% of the total genetic variation was attributable to differences among subpopulations, and 76.22% of the total genetic variation was attributable to differences among accessions within populations (Table 2). The fixation index (*F*_ST_) was 0.24 (*p* < 0.001), suggesting significant high genetic differentiation between the subpopulations.

### 3.5. Population Phylogenetic Relationship

To understand relationships among 82 *R. apiculata* accessions, a maximum likelihood (ML) tree was conducted (Figure 5). Bootstrap values at all branches are high (bootstrap value ≥ 82; mostly equal to 100), indicating that the ML tree was highly reliable. The ML tree shows that 82 *R. apiculata* accessions are divided into two main clusters. Forty-seven accessions from the Gulf of Thailand coast are comprised in cluster I (as shown in the yellow clades in Figure 5), and thirty-four accessions (28 accessions from the Andaman Sea coast and six accessions from the Gulf of Thailand coast) are comprised in cluster II (as shown in the blue clades in Figure 5). Nonetheless, an accession (SNI 04) collected in the Surat Thani province is not in the two clusters of the ML tree. The branching structure of the ML tree is similar to STRUCTURE analysis, in which cluster I accessions are labeled in red text (as shown in orange for group 1 in the STRUCTURE analysis; Figure 4B), whereas cluster II accessions are labeled in blue text (as shown in blue for group 2 in the STRUCTURE analysis; Figure 4B) (Figure 5). In addition, cluster I and II of the ML tree are concordant with the PCA analysis that most of the accessions are divided into two groups: the Gulf of Thailand and Andaman Sea with 67.60% of PC1 (Figure 4C).

## 4. Discussion

Because of the loss of mangrove forest areas, an understanding of *R. apiculata* genetic diversity and population structure is necessary for mangrove management strategies. In the present study, 82 *R. apiculata* accessions (Rhizophoraceae) collected from 17 natural mangrove forests in Thailand were evaluated using SNP markers. Gene diversity (GD) and polymorphic information content (PIC) values have been used to evaluate the level of genetic variation in populations [63]. Based on 2640 SNPs with the MAF values of ≥5%, GD values ranged from 0.16 to 0.50 and PIC values ranged from 0.15 to 0.38, representing moderate to high levels of genetic variation in *R. apiculata*. In addition, the PIC value is commonly used to measure the informativeness of genetic markers [63]. Following the criteria of Botstein et al. [63], 77% (2027 SNPs) of the SNPs were observed to be highly informative markers (0.5 > PIC > 0.25) and 23% (613 SNPs) of the SNPs were less informative markers (PIC < 0.25) (Figure 3; Appendix A). 

To examine the population structure of 82 *R. apiculata* accessions, the Bayesian model-based clustering method and PCA were carried out. Using the STRUCTURE software, the result is shown for the best K = 2, revealing two subpopulations that are divided along the Malay Peninsula between the Gulf of Thailand and the Andaman Sea coasts. In addition, PC1 and PC2 separated the *R. apiculata* accessions into the Gulf of Thailand and the Andaman Sea coasts. These results are concordant with a previous study that reported two groups of *R. apiculata* between the Gulf of Thailand (Bangkok and Surat Thani) and Andaman Sea (Trang) [23]. In contrast, the population of *R. mucronata* in Thailand, which is a sympatric mangrove species with *R. apiculata*, was not clustered into two groups between the Gulf of Thailand and Andaman Sea [23]. Hence, *R. apiculata* in Thailand were shown to independently adapt to their own environments, probably due to the Malay Peninsula barrier that prevents the movement among *R. apiculata* subpopulations between the Gulf of Thailand and Andaman Sea coasts. Based on SNP markers, 63 accessions of *Bruguiera parviflora* (a mangrove species in the family Rhizophoraceae) were also clustered in two subpopulations, the Gulf of Thailand (Surat Thani and Trat) and the Andaman Sea (Phang-nga, Ranong, and Satun) [32]. Furthermore, the population structure of *Ceriops tagal* (one of the mangrove species in the family Rhizophoraceae) was evaluated in Thailand and China and it was shown that populations from the eastern coastline of Thailand were more genetically similar to populations from the South China Sea coast than to populations from the western coastline of Thailand, using inter-simple sequence repeat [64]. The population structure of *R. apiculata* and other mangrove species in the family Rhizophoraceae is conceptually consistent with the land barrier hypothesis of the Malay Peninsula [23].

Interestingly, genetic admixture was found in several accessions in both subpopulations in the Gulf of Thailand (CCO, CMP, SNI, NST, PTN and NWT) and Andaman Sea (STN, TRG, PKT, PNA and RNG) (Figure 4B), which was concordant with previous studies that reported the genetic admixture of *R. apiculata* in Thailand (such as *R. apiculata* populations in Krabi, Phuket, and Ranong), in Indonesia, and in Malayia as well as the genetic admixture of relative *Rhizophora* species in some parts of the IWP region [26,28]. The genetic admixture of *R. apiculata* in Thailand reflected higher levels of genetic diversity with admixed alleles from the two subpopulations.

Genetic diversity and genetic differentiation in the *R. apiculata* population in Thailand were evaluated. Moderate genetic diversity (mean *H*e = 0.39) was observed in the *R. apiculata* population along coastlines in Thailand. This is concordant with the study of *R. apiculata* populations in the Hainan Island, Gulf of Thailand, and the west coast of Thailand [27] as well as the breeding system of *R. apiculata* by mixed mating or predominantly outcrossing, which maintains the level of genetic diversity [26]. In general, both natural and environmental factors, such as mating systems, habitat fragmentation, climate change, and anthropogenic activities, affect the genetic diversity of mangrove species [28,29]. Low genetic diversity was reported in numerous studies on mangrove species, particularly the *Rhizophora* species [24,25,26,28,37,64,65,66]. Furthermore, a high degree of genetic differentiation among two subpopulations (*F*_ST_ = 0.24, *p* < 0.001) was similarly observed in several studies of *R. apiculata* populations, such as in the Strait of Malacca between the east coast (Hainan Island and Gulf of Thailand) and the west coast (Gulf of Thailand) of the Indo-Malayan region (*F*_ST_ = 0.48 (SNPs)) [27], in Thailand between the Bangkok and Trang provinces (*F*_ST_ = 0.875 (five nuclear genes) and *F*_ST_ = 0.688 (two cpDNA regions)) [23], in the Greater Sunda Islands of Indonesia (*F*_ST_ = 0.381 (five microsatellite markers)) [24], in the Indo-Malaysian region between Thailand’s western coast and other regions (*F*_ST_ = 0.242–0.532 (SNPs)) [25], and in Malaysia (*F*_ST_ = 0.315 (three nuclear microsatellite markers)) [28].

The ML tree showed that 81 *R. apiculata* accessions were grouped into two genetic clusters (the Gulf of Thailand and the Andaman Sea) corresponding to the two subpopulations of the STRUCTURE and PCA clustering. Our tree topology is consistent with others [28,66]. For example, the phylogenetic tree by UPGMA (unweighted pair group method with arithmetic mean) revealed the two clusters of *R. apiculata* between the western and eastern regions of Peninsular Malaysia based on microsatellite markers [28]. Using nuclear and chloroplast regions, the NJ (Neighbor-joining) tree of *R. apiculata* populations in the Malay Peninsula regions showed two clusters, the west and south of the Malay Peninsula, and the east of the Malay Peninsula [66]. These support the existence of two genetic patterns in the *R. apiculata* population in Thailand that were geographically isolated on the Gulf of Thailand coast on the east and the Andaman Sea coast on the west. In addition, one accession (SNI 04) from Surat Thani province is separated between cluster I and II based on the ML tree. The STRUCTURE analysis has shown genetic admixture in this accession (Figure 4B). SNI 04 is in the middle of the PCA plot, no cluster with other accessions (Figure 4C). These results suggest that the accession is an admixed individual that combines genetic variation from two genetically differentiated source subpopulations (Gulf of Thailand and Andaman Sea), leading to an increase in the genetic variation within the population of *R. apiculata* in Thailand.

## 5. Conclusions

This study was addressed to examine the genetic diversity and population structure of *R. apiculata* (Rhizophoraceae) along coastlines in Thailand based on SNP markers. Moderate genetic diversity and high genetic differentiation were observed. The results showed two subpopulation differentiations, indicating genetic discontinuity between the coast of the Gulf of Thailand and the Andaman Sea. An AMOVA indicating 76% of variation found within populations and 24% of variation found among populations corroborated these results. Genetic diversity after the divergence of the *R. apiculata* populations might have been caused by geographic isolation.

## Figures and Tables

**Figure 1 biology-11-01449-f001:**
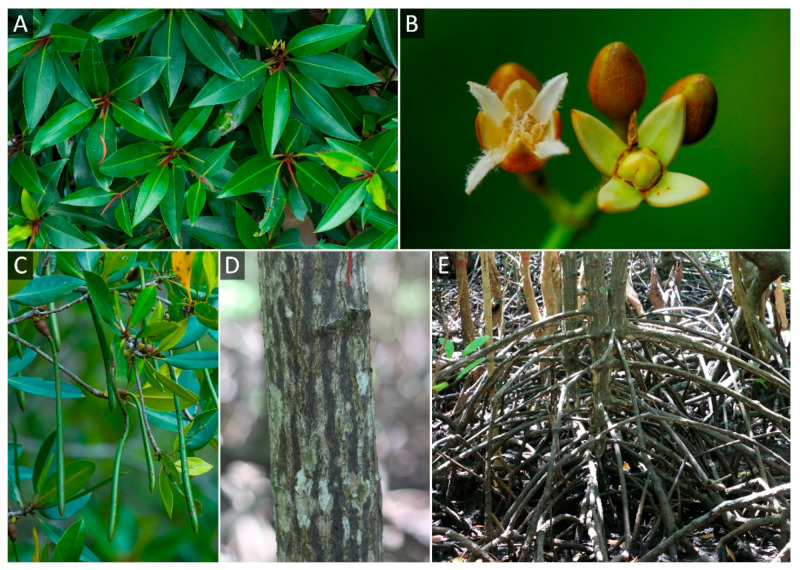
Morphology of *R. apiculata* in Thailand. (**A**) Leaves. (**B**) Flowers. (**C**) Fruits. (**D**) Bark. (**E**) Aerial roots.

**Figure 2 biology-11-01449-f002:**
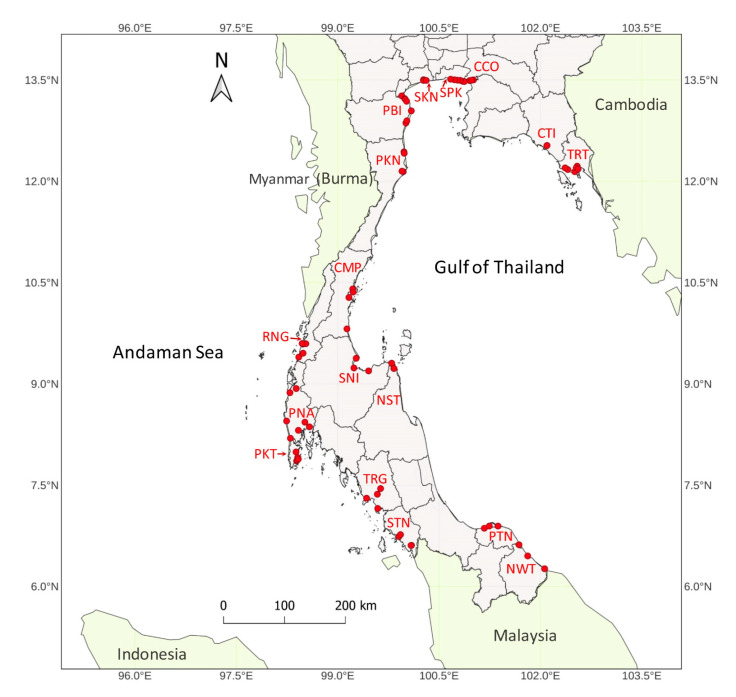
The geographical location of 82 *R. apiculata* accessions in Thailand. Collection sites in 17 provinces are Chachoengsao (CCO), Chumphon (CMP), Chanthaburi (CTI), Nakhon Si Thammarat (NST), Narathiwat (NWT), Phetchaburi (PBI), Prachuap Khiri Khan (PKN), Phuket (PKT), Phang-nga (PNA), Pattani (PTN), Ranong (RNG), Samut Sakhon (SKN), Surat Thani (SNI), Samut Prakan (SPK), Satun (STN), Trang (TRG), and Trat (TRT). Red dots indicate collection sites at which the accessions were collected.

**Figure 3 biology-11-01449-f003:**
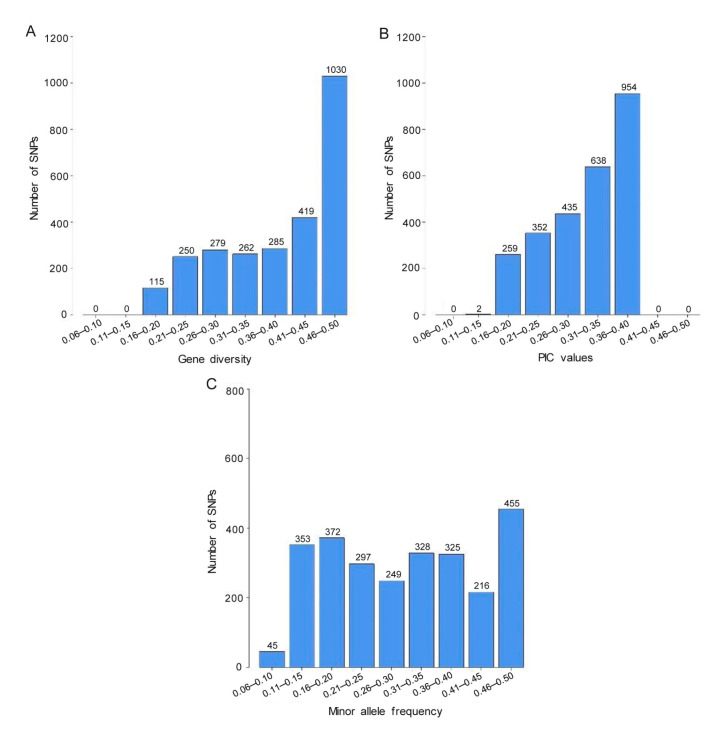
Distribution of (**A**) genetic diversity, (**B**) polymorphism information content (PIC) value, and (**C**) minor allele frequency for 2640 SNPs in 82 *R. apiculata* accessions.

**Figure 4 biology-11-01449-f004:**
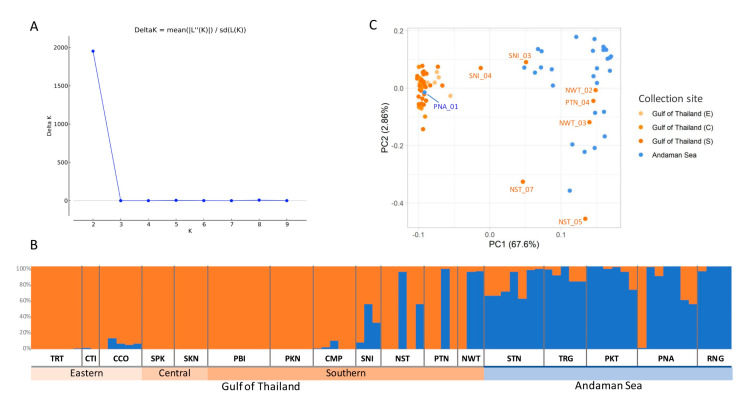
Clustering results of 82 *R. apiculata* accessions. (**A**) The mean estimated delta K values for different numbers of populations assumed K ranging from 1 to 10 in the STRUCTURE analysis. (**B**) Structure plot of *R. apiculata* individuals for K = 2. Each vertical bar represents an individual. The analysis shows two genetic clusters of *R. apiculata* on the Gulf of Thailand and the Andaman Sea coasts. Group 1: TRT-CTI-CCO-SPK-SKN-PBI-PKN-CMP-SNI-NST-PTN-NWT and Group 2: STN-TRG-PKT-PNA-RNG. (**C**) PCA of 82 *R. apiculata* accessions. Each point represents a single individual. Different colors denote four collection sites, which are light orange for the eastern Gulf of Thailand (E), orange for the central Gulf of Thailand (C), dark orange for the southern Gulf of Thailand (S), and blue for the Andaman Sea. The percent of PC1 and PC2 are identified by the *x*-axis and *y*-axis, respectively.

**Figure 5 biology-11-01449-f005:**
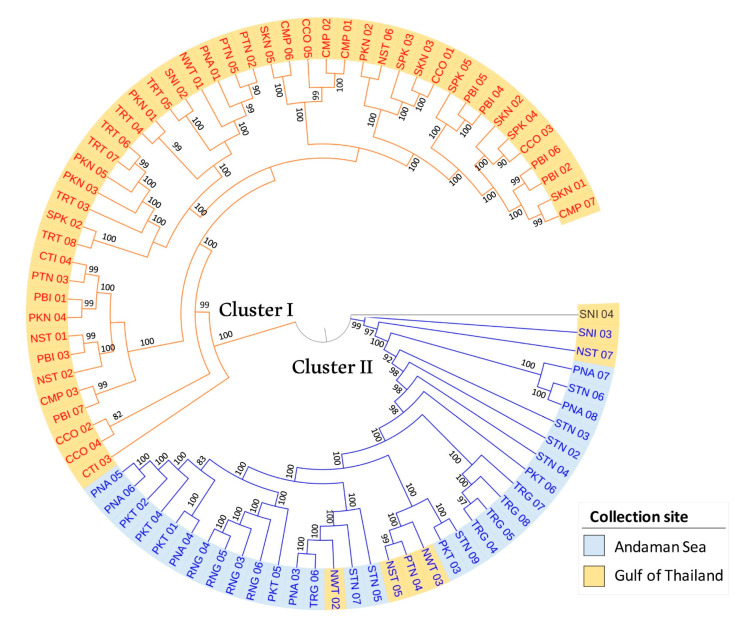
Maximum likelihood dendrogram of 82 *R. apiculata* accessions based on 2640 SNPs. Numbers at the nodes are bootstrap values estimated by permutation test with 1000 replicates. The accessions labeled in red and blue texts represent the first (cluster I) and second (cluster II) groups of *R. apiculata* according to the STRUCTURE analysis, respectively. Blue and yellow highlights indicate collection sites at the Andaman Sea and the Gulf of Thailand, respectively.

**Table 1 biology-11-01449-t001:** Genetic diversity parameters of two genetic clusters of 82 *R. apiculata* accessions in Thailand based on 2640 SNPs.

Population	N	*N*e	*I*	*H*o	*H*e	PPL	*F*
The Gulf of Thailand	53	1.622 ± 0.007	0.553 ± 0.004	0.456 ± 0.006	0.351 ± 0.003	99.92%	−0.140 ± 0.010
The Andaman Sea	29	1.662 ± 0.007	0.573 ± 0.004	0.500 ± 0.006	0.370 ± 0.003	99.05%	−0.258 ± 0.008
Overall	82	1.642 ± 0.005	0.563 ± 0.003	0.478 ± 0.004	0.360 ± 0.002	99.49%	−0.199 ± 0.007

Notes. N: number of samples; *N*e: number of effective alleles; *I*: Shannon’s information index; *H*o: observed. heterozygosity; *H*e: expected heterozygosity; PPL: percentage of polymorphic loci; *F*: inbreeding coefficient.

**Table 2 biology-11-01449-t002:** Analysis of molecular variance (AMOVA) of 82 *R. apiculata* accessions in Thailand.

Source of Variation	df	Sum of Squares	Variance Components	Percentage of Variation	*F*-Statistics
Among populations	1	10768.22	137.70	23.67	*F*_ST_ = 0.24 ***
Within populations	162	71948.59	444.13	76.33	
Total	163	82716.81	581.83		

Notes. degree of freedom (df), *** statistically significant (*p* < 0.001).

## Data Availability

The genome sequence of *R. apiculata* was submitted to the National Center for Biotechnology Information (NCBI), the project accession number was PRJNA846534.

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
