# Peer review of "Assessment of the Genetic Diversity and Population Structure of Rhizophora apiculata Blume (Rhizophoraceae) in Thailand"

_biology, 2022, doi:10.3390/biology11101449_

Round 1

Reviewer 1 Report

Ruang-areerate et al de novo assembled the genome of Rhizophora apiculata, which is a common mangrove species in the Indo-West Pacific region. They further genotyped 82 R. apiculata accessions collected from Thailand using MIG sequencing strategy. Based on the 2640 SNPs, they inferred that R. apiculata populations in Thailand have differentiated into two subpopuplations, corresponding to the geographical regions of Gulf of Thailand and Andaman Sea. They revealed moderate to high levels of genetic diversity in these populations. The data collected in this study is valuable, and the findings are consistent with previous studies. I have two major concerns. First, a high-quality reference genome of R. apiculata had already been published, the authors also used the public reference genome in scaffolding. So, what is the motivation the authors sequenced and assembled a reference genome again? Second, as the authors described in the introduction, the deep genetic differentiation between Gulf of Thailand and Andaman Sea has been reported in a number of previous studies. What is the novelty the authors investigated the populations in Thailand, though the sampling is much more comprehensive across the range of Thailand coasts. Some minor concerns are listed below.

1.   The citations are problematic. For example, at line 84-89, the contents are not consistent with the cited papers. Please check all the citations thoroughly to ensure that all citations match the contents.

2.   Line 136-138. The description of RAD sequencing is too rough. How are the libraries constructed? What technology and platform are used to sequence the libraries? What is done to control quality of the reads?

3.   Line 214-216. It is surprising that only 2640 SNPs were retained from as many as 1,745,767 raw SNPs. It is unclear what criteria were used to filter SNPs? The quality of SNP calling is essential for subsequent population genetic analyses.

4.   Discussion. Obvious admixture was observed in both subpopulations. I suggest the authors to add some discussions on the admixture.

Author Response

We thank the reviewer for your valuable comments on the manuscript and have edited the manuscript to address their concerns.

Reviewer 2 Report

Dear Authors,

The manuscript is very interesting, it explained the genetic diversity and population structure analysis of Rhizhopora apiculata of 82 accessions from natural mangrove forests in 17 provinces of Thailand. There are  some corrections in the manuscript:

Line 90 : what the IWP stand for

Line 96 : The evaluation of genetic diversity and population structure of R. apiculata in Thailand has been limited. Actually you have already explained that there was a study in Line 80-83 : reference [23]. What the differences  this study with [23].

Line 112: Morphology of R. apiculata in Thailand as reference genom? Please describe the general character of R. apiculata sample from 82 accessions

Line 118-124; 128-131: Figure 2. Collection site in the map use number but in the manuscript use abbreviation of the location. Both should the same.  Please rewrite.

Line 128-131 : Figure 2. The geographical location of 82 R. apiculata accessions in Thailand. Collection sites in 17 Provinces à please explain that dots/points represent the accession.

Line 206-208: The size of our R. apiculata genome (231 Mb) is similar to the size of the previously reported R. apiculata  genomes (232 Mb) [37,62], à where the source of the samples?

Line 242: accessions were divided into two groups: the Gulf of Thailand and the Andaman Sea, could you write the collection site for those locations?

Line 276: Did the result of Population phylogenetic relationship give strong explanation of previous PCA analyses result?

Line 320: B. parviflora, what B stand for?

Line 349-350 : The result showed that R. apiculata accessions were grouped into two genetic  clusters (the Gulf of Thailand and the Andaman Sea) corresponding to the two subpopulations and it caused by reproduction and geographic isolation, please look at Figure 2, how is the genetic diversity among those in cluster of the Gulf of Thailand,  for example the comparison of the sample from location 1 and 2 and 8,9,10.

Please revise the manuscript.

Thank you.

Author Response

We thank for your valuable comments on the manuscript and have edited the manuscript to address their concerns.

Reviewer 3 Report

This paper used 10x genomics technology to obtain a reference whole-genome sequence of Rhizophora apiculata, and assessed the genetic diversity and population structure of 82 R. apiculata accessions in Thailand based on 2,640 single nucleotide polymorphisms (SNPs). The results revealed a moderate genetic diversity and high genetic differentiation between populations. Genetic structure was affected by geographic isolation. The methods and analysis is sound. 

Specific comments are listed below on a line by line basis.

1. Line 36-38: The cause of genetic differentiation (geographic isolation by the Malay Peninsula) should be added in abstract.

2. Fig.2: Latitude and longitude information should be added. What do the red dots represent?

3. Table1: “Cluster 1 and Cluster 2should change to “the Gulf of Thailand and the Andaman Sea”.

4. Line 179: The subtitle is not appropriate, please reconsider. Change subtitle number to 2.6.

5. Line 188: The subtitle number should change to 2.7.

6. Line 257-262: What does “PPL” was mentioned in Table1 explain?

7. Table5: Cluster I accessions are labeled in red text (Gulf of Thailand), and cluster II accessions are labeled in blue text (Andaman Sea). However, the SNI 04 was not associated with these two clusters but labeled in red text. Please check it.

8. Line 313: the two largest PC” should be interpreted as PC1 and PC2.

9. Line 349-359: An accession (SNI 04) which was not associated with two clusters is ignored in the discussion about ML tree

10. Line 367-369: This statement, especially “reproduction”, is true but not relevant to be the conclusion of this study. Please add more references for support in Discussion.

Author Response

(The authors gave the same response as above.)

Reviewer 4 Report

Manuscript ID: biology-1821729

In this ms entitled “Assessment of the genetic diversity and population structure of Rhizophora apiculata Blume (Rhizophoraceae) in Thailand”, the Authors characterized the genetic diversity and population structure of 82 accessions of mangrove species Rhizophora apiculata by performing sequence analysis. In these R. apiculata genomes, 2,640 SNPs were identified and genetic diversity was observed among and within mangrove populations. This study provided useful information for the management of the mangrove forests in Thailand.

This work is very interesting, but the results were not clearly described, so the paper has to be revised in some parts. This reviewer recommends minor revisions.

Here are some specific comments:

 ABSTRACT

L20-21: check repetitions, merge these two sentences

INTRO

L66-67: use abbreviation “R.” for “Rhizophora

L71: add “(AFLP)” after “polymorphisms”

L74-78: check references [24] and [25]; add ref. [35] for RAD-sequencing

M&M

L136: in short, report here the details of 10X Genomics tech.

L137-138: here, report the details of MGI tech.

RESULTS

L231: check English language, change “identified” to “performed”

L259: these data are shown in Table 1, it is not necessary to report them here, please remove them

L284-285: check English language, and precise where is located the accession SNI 04

DISCUSSION

Here the results are reported again, please reduce the space of results for the benefit of a broader and detailed discussion.

L296-299: check repetitions, remove “genetic” in L297, and “the genetic diversity and population structure of” in L299

L316: add results for R. mucronata from [23]

L320: check reference [24] for B. parviflora

L327: remove “Obviously,”

L358-359: what did you mean for “discrete”, please explain

CONCLUSIONS

L361: check English language, change “is” to “was addressed” or equivalent verb

L3636-364: remove “of population structure, PCA, and phylogenetic analyses”

L367: remove “Moderate”

L368: use plural form for “population”

L368: change “and” to “after their”

Author Response

(The authors gave the same response as above.)
